# OpenIllumination: A Multi-Illumination Dataset for Inverse Rendering Evaluation on Real Objects

**Isabella Liu[1*], Linghao Chen[2*], Ziyang Fu[1], Liwen Wu[1], Haian Jin[2], Zhong Li[3],**

**Chin Ming Ryan Wong[3], Yi Xu[3], Ravi Ramamoorthi[1], Zexiang Xu[4], Hao Su[1]**

[1]UC San Diego, [2]Zhejiang University, [3] OPPO US Research Center,
[4] Adobe Research, [*]Equal Contribution,
{lal005,zifu,liw026,ravi,haosu}@ucsd.edu
{chenlinghao,haian}@zju.edu.cn
zexiangxu@gmail.com
{zhong.li,ryan.wong,yi.xu}@oppo.com

## Abstract

We introduce OpenIllumination, a real-world dataset containing over 108K images of 64 objects with diverse materials, captured under 72 camera views and a large number of different illuminations. For each image in the dataset, we provide accurate camera parameters, illumination ground truth, and foreground segmentation masks. Our dataset enables the quantitative evaluation of most inverse rendering and material decomposition methods for real objects. We examine several state-of-the-art inverse rendering methods on our dataset and compare their performances. The dataset and code can be found on the project page: https://oppo-us-research.github.io/OpenIllumination.

## 1 Introduction

Recovering object geometry, material, and lighting from images is a crucial task for various applications, such as image relighting and view synthesis. While recent works have shown promising results by using a differentiable renderer to optimize these parameters using the photometric loss [51, 53, 52, 20, 32], they can only perform a quantitative evaluation on synthetic datasets since it is easy to obtain ground-truth information. In contrast, they can only show qualitative results instead of providing quantitative evaluations in real scenes.

Nevertheless, it is crucial to acknowledge the inherent gap between synthetic and real-world data, for real-world scenes exhibit intricate complexities, such as natural illuminations, diverse materials, and complex geometry, which may present challenges that synthetic data fails to model accurately. Consequently, it becomes imperative to complement synthetic evaluation with real-world data to validate and assess the ability of inverse rendering algorithms in practical settings.

It is highly challenging to capture real objects in practice. A common approach to capturing real-world data is using a handheld camera [20, 53]. Unfortunately, this approach frequently introduces the occlusion of ambient light by photographers and cameras, consequently resulting in different illuminations for each photograph. Such discrepancies are unreasonable for most methods that assume a single constant illumination. Furthermore, capturing images under multiple illuminations with a handheld camera often produces images with highly different appearances and results in inaccurate and even fail camera pose estimation, particularly for feature matching-based methods such as COLMAP [37]. Recent efforts have introduced some datasets [33, 43, 21] that incorporate

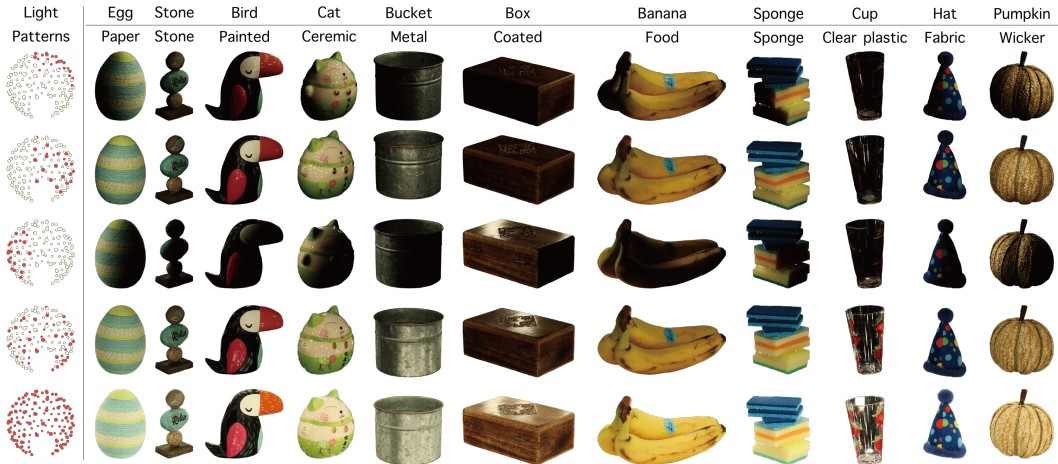

Figure 1: **Some example images in the proposed dataset.** The dataset contains images of various objects with diverse materials, captured under different views and illuminations. The leftmost column visualizes several different illumination patterns, with **red** and yellow indicating activated and deactivated lights. The name and material for each object are listed in the first and second rows. The materials are selected from the OpenSurfaces [3] dataset.

multiple illuminations in real-world settings. However, as shown in Tab. 1, most of them are limited either in the number of views [33, 21] or the number of illuminations [21]; few of them provide object-level data as well. Consequently, these existing datasets prove unsuitable for evaluating inverse rendering methods on real-world objects.

To address this, we present a new dataset containing objects with a variety of materials, captured under multiple views and illuminations, allowing for reliable evaluation of various inverse rendering tasks with real data. Our dataset was acquired using a setup similar to a traditional light stage [10, 11], where densely distributed cameras and controllable lights are attached to a static frame around a central platform. In contrast to handheld capture, this setup allows us to precisely pre-calibrate all cameras with carefully designed calibration patterns and reuse the same camera parameters for all the target objects, leading to not only high calibration accuracy but also a consistent evaluation process (with the same camera parameters) for all the scenes.

On the other hand, the equipped multiple controllable lights enable us to flexibly illuminate objects with a large number of complex lighting patterns, facilitating the acquisition of illumination ground truth.

| Dataset | Capturing device | Lighting condition | Number of illuminations | HDR | Number of scenes/objects | Number of views |
|---|---|---|---|---|---|---|
| DTU [19] | gantry | pattern | 7 | ✗ | 80 scenes | 49/64 |
| NeRF-OSR [36] | commodity camera | env | 5∼11 | ✗ | 9 scenes | ∼360 |
| DiLiGenT [39] | commodity camera | OLAT | 96 | ✓ | 10 objects | 1 |
| DiLiGenT-MV [26] | studio/desktop scanner | OLAT | 96 | ✓ | 5 objects | 20 |
| NeROIC [23] | commodity camera | env | 4∼6 | ✗ | 3 objects | 40 |
| MIT-Intrinsic [15] | commodity camera | OLAT | 10 | ✗ | 20 objects | 1 |
| Murmann et al. [33] | light probe | env | 25 | ✗ | 1000 scenes | 1 |
| LSMI [21] | light probe | env | 3 | ✗ | 2700 scenes | 1 |
| ReNe [43] | gantry | OLAT | 40 | ✗ | 20 objects | 50 |
| Ours | light stage | pattern+OLAT | 13 pattern+ 142 OLAT | ✓ | 64 objects | 72 |

Table 1: **Comparison between representative multi-illumination real-world datasets.** Env. stands for environment lights.

With the help of high-speed cameras running at 30 fps, we are able to capture OLAT (One-Light-At-a-Time) images with a very high efficiency, which is critical for capturing data at a large scale. In the end, we have captured over 108K images, each with a well-calibrated camera and illumination

parameters. Moreover, we also provide high-quality object segmentation masks by designing an efficient semi-automatic mask labeling method.

We conduct baseline experiments on several tasks: (1) joint geometry-material-illumination estimation; (2) joint geometry-material estimation under known illumination; (3) photometric stereo reconstruction; (4) Novel view synthesis to showcase the ability to evaluate real objects on our dataset. To the best of our knowledge, by the time of this paper's submission, there are no other real datasets that can be used to perform the quantitative evaluation for relighting on real data.

In summary, our contributions are as follows:

- We capture over 108K images for real objects with diverse materials under multiple viewpoints and illuminations, which enables a more comprehensive analysis for inverse rendering tasks across various material types.

- The proposed dataset provides precise camera calibrations, lighting ground truth and accurate object segmentation masks.

- We evaluate and compare the performance of multiple state-of-the-art (SOTA) inverse rendering and novel view synthesis methods. We perform quantitive evaluation of relighting real object under unseen illuminations.

## 2 Related works

**Inverse rendering.** Inverse rendering has been a long-standing task in the fields of computer vision and graphics, which focuses on reconstructing shapes and materials from multi-view 2D images. A great amount of work [5, 14, 18, 25, 47, 34, 52, 54] has been proposed for this task. Some of them make use of learned domain-specific priors [5, 12, 2, 27]. Some other works rely on controllable capture settings to estimate the geometry and material, such as structure light [48], circular LED lights [55], collocated camera and flashlight [50, 5, 4], and so on.

Recently, a lot of works use neural representations to support inverse rendering reconstruction under unknown natural lighting conditions [20, 6, 52, 54, 7, 32, 51]. By combining the popular neural representations such as NeRF [30] or SDF [45, 49] with physically-based rendering model [8], they can achieve shape and reflectance reconstruction with image loss constraint. Although these works can achieve high-quality reconstruction, they can only evaluate relighting performance under novel illumination on synthetic data because of the lack of high-quality real object datasets.

**Multi-illumination datasets.** Multi-illumination observations intuitively provide more cues for computer vision and graphics tasks like inverse rendering. Some works have utilized the temporal variation of natural illumination, such as sunlight and outdoor lighting. These "in-the-wild" images are typically captured using web cameras [46, 41, 36] or using controlled camera setups [40, 24]. Another line of work focuses on indoor scenes, while indoor scenes generally lack a readily available source of illumination that exhibits significant variation. In this case, a common approach involves using flash and no-flash pairs [35, 13, 1]. Applications like denoising, mixed-lighting white balance, and BRDF capture benefits from these kinds of datasets. However, other applications like photometric stereo and inverse rendering usually require more than two images and more lighting conditions for reliable results, which these datasets often fail to provide.

## 3 Dataset construction

### 3.1 Dataset overview

The OpenIllumination dataset contains over 108K images of 64 objects with diverse materials. Each object is captured by 48 DSLR cameras under 13 lighting patterns. Additionally, 20 objects are captured by 24 high-speed cameras under 142 OLAT setting.

Fig. 1 shows some images captured under different lighting patterns, while the images captured under OLAT illumination can be found in Fig. 5.

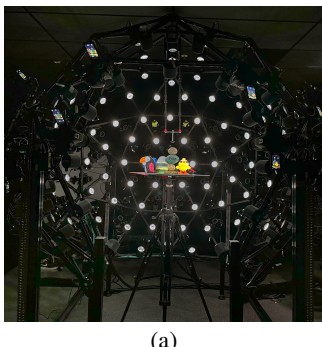
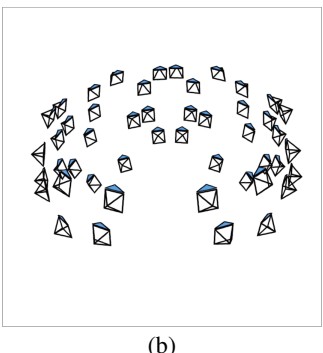
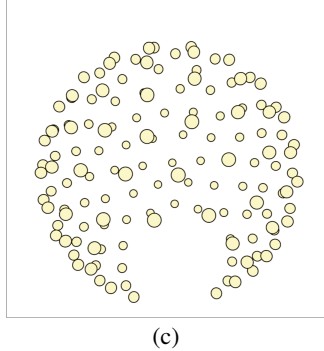

|(a)|(b)|(c)|
|---|---|---|

Figure 2: (a) The capturing system contains 48 DSLR cameras (Canon EOS Rebel SL3), 24 high-speed cameras (HR-12000SC), and 142 controllable linear polarized LED. (b) The calibrated DSLR camera poses. (c) The reconstructed light positions.

Our dataset includes a total of 24 diverse material categories, such as plastic, glass, fabric, ceramic, and more. Note that one object may possess several different materials, thus the number of materials is larger than the number of objects.

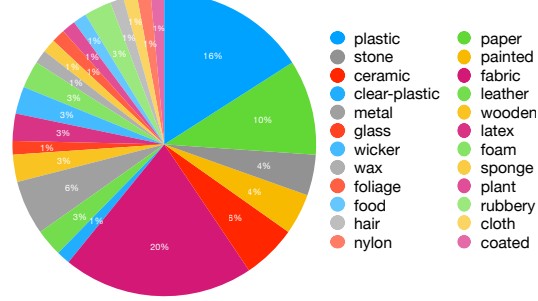

## 3.2 Camera calibration

The accuracy of camera calibration highly affects the performance of most novel view synthesis and inverse rendering methods. Previous works [20, 53] typically capture images by handheld cameras and employ COLMAP [37] to estimate camera parameters. However, this approach heavily relies on the object's textural properties, which is challenging in instances where the object lacks texture or exhibits specular reflections from certain viewpoints. These challenges can obstruct accurate feature matching, consequently reducing the precision of camera parameter estimation. Ultimately, the reliability of inverse rendering outcomes is undermined, and finding out whether inaccuracies are caused by erroneous camera parameters or limitations of the inverse rendering method itself becomes a challenging problem. Leveraging the capabilities of our light stage, wherein camera intrinsics and extrinsic can be fixed when capturing different objects, we employ COLMAP to recover the camera parameters on a rich textured and low-specularity scene. For each subsequently captured object, we use this set of camera parameters instead of performing recalibration. The results of camera calibration are visualized in Fig. 2(b).

## 3.3 Light calibration

In this section, we propose a chrome-ball-based lighting calibration method to obtain the ground-truth illumination which plays a critical role in the relighting evaluation.

Our data are captured in a dark room where a set of linear polarized LEDs are placed on a sphere uniformly as the only lighting source. Each light can be approximated by a Spherical Gaussian (SG), defined as the following form [44]:

$$G(\nu; \boldsymbol{\xi}, \lambda, \boldsymbol{\mu}) = \boldsymbol{\mu}\, e^{\lambda(\nu \cdot \boldsymbol{\xi} - 1)}, \qquad (1)$$

where $\nu \in \mathbb{S}^2$ is the function input, representing the incident lighting direction to query, $\boldsymbol{\xi} \in \mathbb{S}^2$ is the lobe axis, $\lambda \in \mathbb{R}_+$ is the lobe sharpness, and $\boldsymbol{\mu} \in \mathbb{R}_+^n$ is the lobe amplitude.

We utilize a chrome ball to estimate the 3D position of each light. Assuming the chrome ball is highly specular and isotropic, its position and radius are known, and cameras and lights are evenly distributed around the chrome ball. For each LED single light, at least one camera can capture the reflected light rays out from its starting location. The incident light direction can be computed via:

$$I = -T + 2(I \cdot N)N, \qquad (2)$$

where $I$ is the incident light direction that goes out from the point of incidence, $N$ is the normal of the intersection point on the surface, and $T$ is the direction of the reflected light.

For each LED light, its point of incidence on the chrome ball can be captured by multiple cameras, and for each camera $i$, we can compute an incident light direction $I_i$, which should have the least distance from the LED light location $p$. Therefore, to leverage information from multiple camera viewpoints, we seek to minimize the sum of distances between the light position and incident light directions across different camera views. This optimization is expressed as:

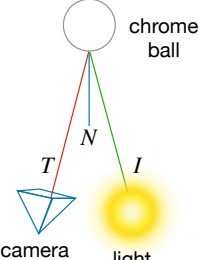

$$L(p) = \sum_i d(p, I_i), \|p\| = 1, \qquad (3)$$

where $p$ represents the light position to be determined, $d(p, I_i)$ denotes the L2 distance between the light and the incident light direction corresponding to view $i$, and the constraint $\|p\| = 1$ ensures that the lights lie on the same spherical surface as the cameras. The reconstructed light distribution, depicted in Fig. 2(c), closely aligns with the real-world distribution.

After estimating the 3D position for each light, we need to determine the lobe size for them. Since the lights in our setup are of the same type, we can estimate a global lobe size for all lights. By taking one OLAT image of the chrome ball as input, we flatten it into an environment map. Subsequently, we optimize the parameters of the Spherical Gaussians (SGs) model to minimize the difference between the computed environment map and the observed environment map. The final fitted lobe size parameter we use is 236.9705.

Since all the lights have identical lighting intensities, and the lighting intensity can be of arbitrary scale because of the scale ambiguity between the material and lighting, we set the lighting intensity to 5 for all lights.

### 3.4   Semi-automatic high-quality mask labeling

To obtain high-quality segmentation masks, we use Segment-Anything [22] (SAM) to perform instance segmentation. However, we find that the performance is not satisfactory. One reason is that the object categories are highly undefined. In this case, even combining the bounding box and point prompts cannot produce satisfactory results. To address this problem, we use multiple bounding-box prompts to perform segmentation for each possible part and then calculate a union of the masks as the final object mask. For objects with very detailed and thin structures, e.g. hair, we use an off-the-shelf background matting method [28] to perform object segmentation.

## 4   Baseline experiments

### 4.1   Inverse rendering evaluation

In this section, we conduct experiments employing various learning-based inverse rendering methods on our dataset. Throughout these experiments, we carefully select 10 objects exhibiting a diverse range of materials, and we partition the images captured by DSLR cameras into training and testing sets, containing 38 and 10 views respectively.

**Baselines.** We validate six recent learning-based inverse rendering approaches assuming single illumination conditions: NeRD [6], Neural-PIL [7], PhySG [51], InvRender [54], nvdiffrec-mc [16], and TensoIR [20]. Moreover, we validate three of them [6, 7, 20] that support multiple illumination optimization.

**Joint geometry-material-illumination estimation.** For experiments under single illumination, we use images captured with all lights activated, while for multi-illumination, we select images taken under three different lighting patterns.

NeRD[6] is observed to exhibit high instability. In many cases, NeRD fails to learn a meaningful environment map. Neural-PIL [7] generates fine environment maps and produces high-quality renderings. However, the generated environment map incorporates the albedo of objects and fails to produce reasonable diffuse results in multi-illumination conditions. Both NeRD and Neural-PIL

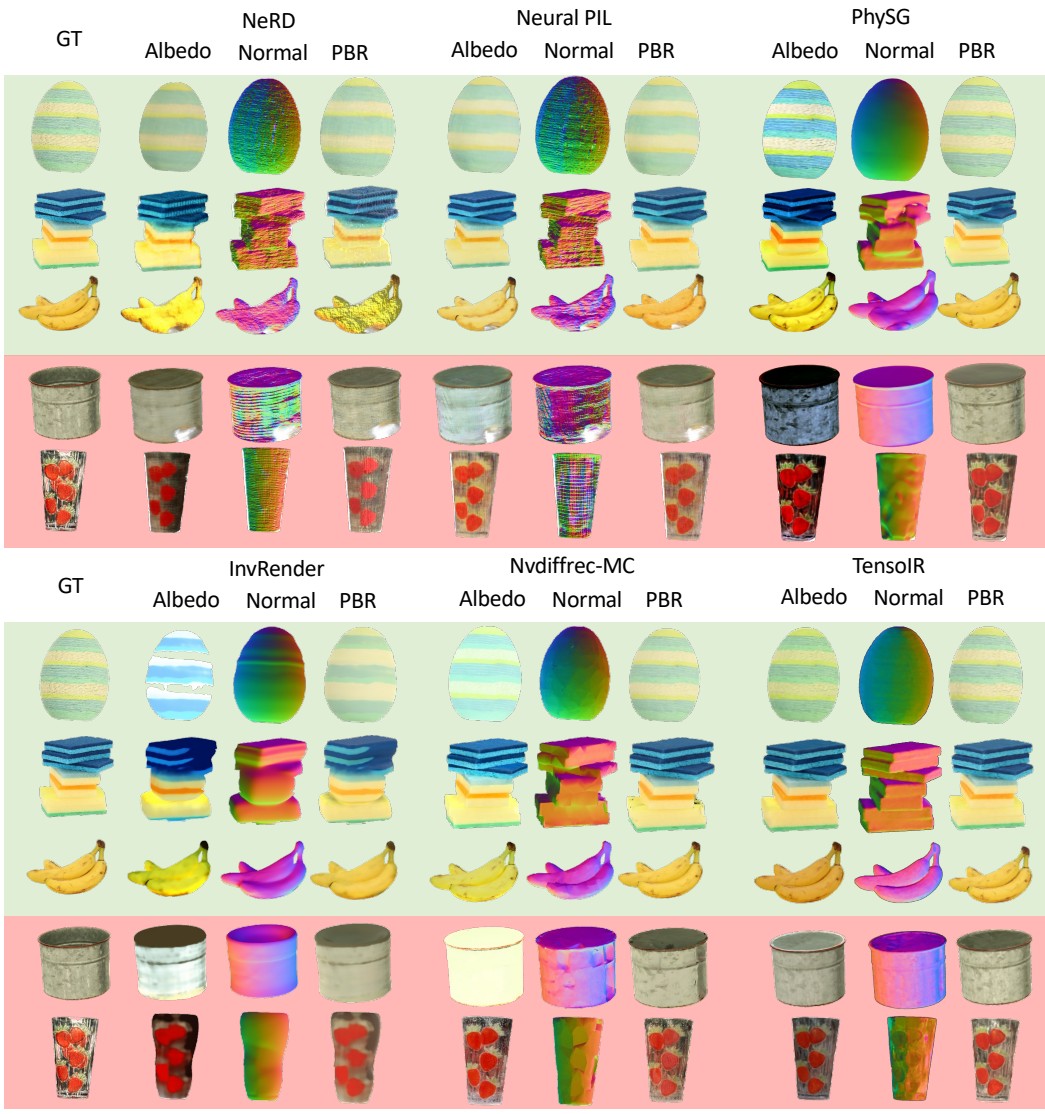

Figure 3: The object reconstruction on our dataset from three inverse rendering baselines under single illumination. Objects highlighted by **green** color are easier tasks in our dataset, while objects in **red** color are more difficult tasks that involve more complicated materials like metal and clear plastic.

suffer from map fractures in roughness, normal, and albedo, providing visible circular cracks, which we attribute to the overfitting of the environment map, where certain colors become embedded within it. PhySG [51] applies specular BRDFs allowing for a better approximate evaluation of light transport. PhySG shows commendable results on metal and coated materials, simulating a few highlights. However its geometry learning was inaccurate, and it performed poorly in objects with multiple specular parts, failing to reproduce any prominent highlights. InvRender [54] models spacially-varying indirection illumination and the visibility of direct illumination. However, its reconstructed geometry tends to lack detail and be over-smooth on some objects. nvdiffrec-mc [16] incorporates Monte Carlo integration and a denoising module during rendering to achieve a more efficient and stable convergence in optimization. It achieves satisfactory relighting results on most objects. But the quality of geometry detail as shown in the reconstructed normal map is affected by the grid resolution of DMTet [38]. TensoIR [20] also exhibits satisfactory performance. However, it still encounters challenges in generating good results for highly specular surfaces, as shown in the fourth row in Fig. 3. Moreover, since TensoIR models materials using a simplified version of Disney BRDF [8], which fixes the $F_0$ in the fresnel term to be 0.04, its representation capabilities are limited, and certain materials such as metal and transparent plastic may not be accurately modeled, as

illustrated in the fifth row in Fig. 3 and Tab. 2, where TensoIR only achieve about 22 PSNR on the translucent plastic cup.

Overall, all the methods struggle with modeling transparency or complex reflectance because of the relatively simple BRDF used in rendering. For concave objects, such as the metal bucket shown in Fig. 3, NeRF-based methods have difficulty learning the correct geometry. In addition, compared to single illumination, two of our baselines, NeRD and NeuralPIL show inferior performance under multi-illumination, and the baseline TensoIR maintains a high quality of the reconstruction.

| Object | egg | stone | bird | box | pumpkin | hat | cup | sponge | banana | bucket |
|---|---|---|---|---|---|---|---|---|---|---|
| Material | paper | stone | painted | coated | wooden | fabric | clear plastic | sponge | food | metal |
| NeRD | 33.40 | 27.20 | 26.81 | 22.80 | 23.81 | 27.64 | 22.06 | 26.78 | 25.54 | 26.14 |
| Neural-PIL | 34.42 | 29.41 | 29.17 | 25.49 | 27.59 | 30.14 | 22.55 | 31.01 | 31.61 | 27.73 |
| PhySG | 35.06 | 30.72 | 29.02 | 26.56 | 27.32 | 31.16 | 21.86 | 30.70 | 34.39 | 29.25 |
| InvRender | 31.52 | 25.51 | 24.96 | 23.80 | 25.43 | 22.79 | 21.62 | 24.20 | 29.34 | 26.18 |
| nvdiffrec-mc | 35.77 | 31.51 | 30.20 | 27.29 | 28.12 | 31.19 | 22.08 | 32.68 | 35.60 | 28.52 |
| TensoIR | 34.88 | 29.96 | 30.21 | 26.80 | 28.20 | 31.96 | 22.13 | 32.49 | 34.77 | 29.32 |

Table 2: **Inverse rendering evaluation results under single illumination.** We validate six inverse rendering baselines with static illumination. We report the PSNR results for each object.

| Object | egg | stone | bird | box | pumpkin | hat | cup | sponge | banana | bucket |
|---|---|---|---|---|---|---|---|---|---|---|
| Material | paper | stone | painted | coated | wooden | fabric | clear plastic | sponge | food | metal |
| NeRD | 26.32 | 24.20 | 24.34 | 21.05 | 18.74 | 23.14 | 21.59 | 17.73 | 21.22 | 16.48 |
| Neural-PIL | 30.84 | 28.48 | 28.47 | 25.45 | 25.74 | 29.80 | **22.44** | 29.41 | 30.59 | 26.06 |
| TensoIR | **34.51** | **29.88** | **30.21** | **26.53** | **27.96** | **31.58** | 22.09 | **31.87** | **34.35** | **28.91** |

Table 3: **Inverse rendering evaluation results under multi-illumination.** We select three light patterns from our dataset to validate three baselines that support multiple illuminations. We report the PSNR results for each object.

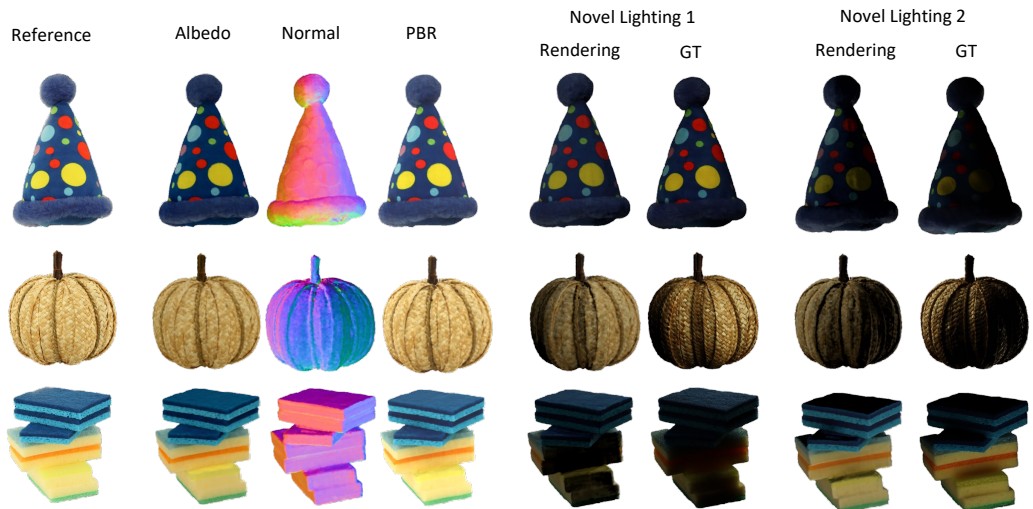

Figure 4: Relighting results of TensoIR under novel illumination. We show the reconstructed albedo, normal, and PBR results. For each novel illumination, we show the rendering and ground-truth captured images.

**Joint geometry-material estimation under known illumination.** As introduced in Sec. 3.1, we capture the objects under different illuminations. For each illumination, we provide illumination

ground truth represented as a combination of Spherical Gaussian functions. This enables us to evaluate the performance of relighting under novel illumination with the decomposed material and geometry.

| Object | egg | stone | bird | box | pumpkin | hat | cup | sponge | banana | bucket |
|--------|-----|-------|------|-----|---------|-----|-----|--------|--------|--------|
| Material | paper | stone | painted | coated | wooden | fabric | clear plastic | sponge | food | metal |
| PSNR | 31.99 | 31.07 | 30.16 | 27.57 | 27.16 | 32.38 | 22.96 | 30.86 | 32.13 | 27.13 |

Table 4: Performance of relighting under novel illumination using TensoIR.

Tab. 4 shows the relighting performance of TensoIR [20] on 10 objects. Fig. 4 shows the material decomposition and the relighting visualizations. In general, TensoIR performs better on diffuse objects than on metal and transparent objects.

## 4.2 Photometric stereo

Photometric stereo (PS) is a well-established technique to reconstruct a 3D surface of an object [18]. The method estimates the shape and recovers surface normals of a scene by utilizing several intensity images obtained under varying illumination conditions with an identical viewpoint [17, 42]. By default, PS assumes a Lambertian surface reflectance, in which normal vectors and image intensities are linearly dependent on each other. During our capturing, we place circular polarizers over each light source, we also place a circular polarizer of the same sense in front of the camera to cancel out the specular reflections [29]. Fig. 5 shows the reconstructed albedo and normal map from the OLAT images in our dataset.

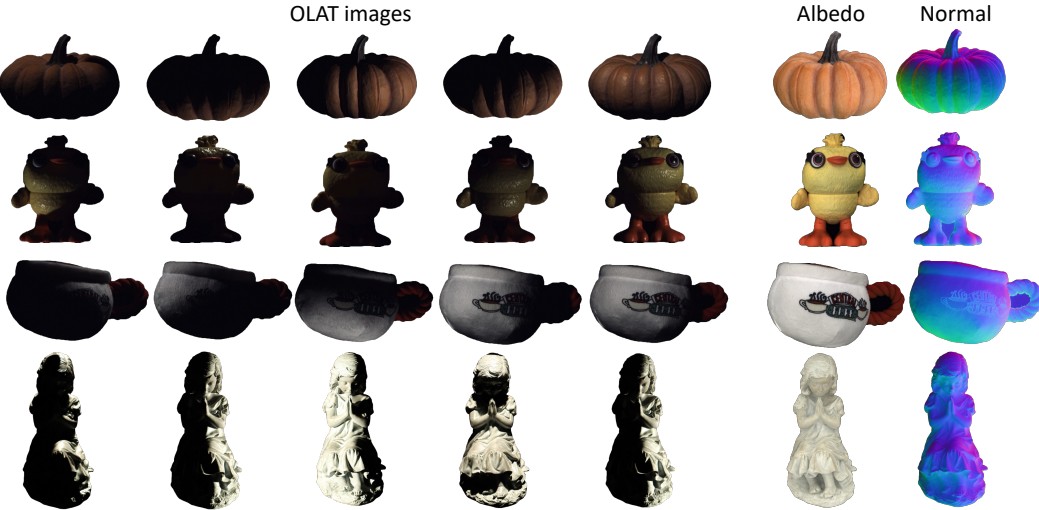

Figure 5: Results of photometric stereo using the OLAT images in our dataset.

## 4.3 Novel view synthesis

While our dataset was primarily proposed for evaluating inverse rendering approaches, the multi-view images in it can also serve as a valuable resource for evaluating novel view synthesis methods. In this section, we perform experiments utilizing several neural radiance field methods to validate the data quality of our dataset. We conduct experiments employing the vanilla NeRF [30], TensoRF [9], Instant-NGP [31], and NeuS [45]. The quantitative results, as presented in Tab. 5, demonstrate the exceptional quality of our data and the precise camera calibration, as evidenced by the consistently high PSNR scores attained.

| Object | egg | stone | bird | box | pumpkin | hat | cup | sponge | banana | bucket |
|--------|-----|-------|------|-----|---------|-----|-----|--------|--------|--------|
| Material | paper | stone | painted | coated | wooden | fabric | clear plastic | sponge | food | metal |
| NeRF [30] | 33.53 | 29.32 | 29.64 | 25.38 | 26.95 | 31.29 | **22.52** | 31.36 | 33.65 | 28.54 |
| TensoRF [9] | 32.42 | 29.84 | 28.45 | **25.49** | 27.54 | 31.50 | 20.87 | 31.34 | 34.32 | 29.28 |
| I-NGP [31] | **34.07** | **30.62** | 29.91 | 25.83 | **27.93** | **32.51** | 22.51 | **32.71** | **34.98** | 29.72 |
| NeuS [45] | 33.43 | 29.78 | **30.00** | 25.47 | 27.83 | 31.93 | 22.13 | 32.44 | 34.17 | **29.99** |

Table 5: **Novel-view-synthesis PSNR on NeRF, TensoRF, Instant-NGP, and NeuS.**

## 4.4 Ablation study

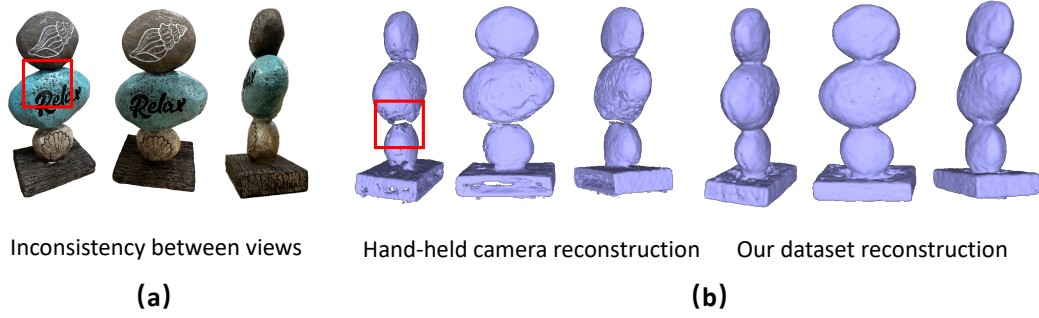

Inconsistency between views

**(a)**

Hand-held camera reconstruction     Our dataset reconstruction

**(b)**

Figure 6: **(a)** Capturing using a handheld camera often introduces inconsistent illuminations. **(b)** Geometry reconstruction using data in our dataset delivers higher completion than using data captured by handheld cameras.

As depicted in Fig. 6 (a), the utilization of handheld cameras in the capture process frequently gives rise to inconsistent illumination between different viewpoints because of the changing occlusion of light caused by the moving photographer, thereby breaching the static illumination assumption for most inverse rendering methods. In Fig. 6 (b), we use a handheld smartphone to capture data under a similar setup in the dome. Experiments on handheld cameras tends to inadequately ensure an extensive range of viewpoints, thereby frequently resulting in the incompleteness of the reconstructed objects. Conversely, our dataset delivers a superior range of viewpoints and maintains consistency across different objects, thereby producing a more complete reconstruction. This demonstrates the high quality of our dataset and establishes its suitability as an evaluation benchmark for real-world objects.

## 5 Limitation

There are several limitations and future directions to our work. **(1)** Since we use the light stage to capture the images in a dark room, the illumination is controlled strictly. Thus there exists a gap between the images in this dataset and in-the-wild captured images. **(2)** Although we use state-of-the-art methods for segmentation, the mask consistency across different views for smaller objects with fine details, such as hair, is not considered yet. **(3)** Due to the limited space, the sizes of the objects in the dataset are restricted to 10∼20 cm, and the cameras are not highly densely distributed.

## 6 Conclusion

In this paper, we introduce a multi-illumination dataset OpenIllumination for inverse rendering evaluation on real objects. This dataset offers crucial components such as precise camera parameters, ground-truth illumination information, and segmentation masks for all the images. OpenIllumination provides a valuable resource for quantitatively evaluating inverse rendering and material decomposition techniques applied to real objects for researchers. By analyzing various state-of-the-art inverse

rendering pipelines using our dataset, we have been able to assess and compare their performance effectively. The release of both the dataset and accompanying code will be made available, encouraging further exploration and advancement in this field.

## 7    Acknowledgement

This work was supported in part by ONR grant N00014-23-1-2526 and NSF grant 2110409. We acknowledge gift support from Adobe, Google, Meta, Qualcomm, Oppo, the Ronald L. Graham Chair and the UC San Diego Center for Visual Computing.

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
