# Supplementary Material for "OpenIllumination: A Multi-Illumination Dataset for Inverse Rendering Evaluation on Real Objects"

## 1 Dataset access

**URL and data cards.** The dataset can be viewed at https://oppo-us-research.github.io/OpenIllumination and downloaded from https://huggingface.co/datasets/OpenIllumination/OpenIllumination.

**Author statement**. We bear all responsibility in case of violation of rights. We confirm the CC BY (Attribution) 4.0 license for this dataset.

**Hosting, licensing, and maintenance plan.** We host the dataset on HuggingFace [2], and we confirm that we will provide the necessary maintenance for this dataset.

**DOI.** 10.57967/hf/1102.

**Structured metadata.** The metadata is at https://huggingface.co/datasets/OpenIllumination/OpenIllumination.

## 2 Capturing details

### 2.1 Object masks

As mentioned in the main paper, our capturing process involves using a device similar to a light stage, which has a diameter of approximately 2 meters. The device consists of cameras and LED lights evenly distributed on the surface of a sphere, all oriented toward the center. To position the object roughly at the center, we utilize two types of supports, as illustrated in Fig. 1(a). However, due to the presence of camera angles that capture views from the bottom to the top, as depicted in Fig. 1(b), certain areas of the surface may be occluded by the supporting device. Consequently, these areas become invisible in these specific views while remaining visible in other views after applying the masking process. This introduces ambiguity to the density field network and leads to inferior performance.

To address this issue and eliminate density ambiguity, we incorporate certain parts of the supporting device in the training images. During the evaluation, we evaluate the PSNR using a separate set of masks that only contain the object. In the dataset, we utilize the *com_mask*, which combines the supporting device and object masks, during the training phase. For inference and evaluation, we employ the *obj_mask*, which represents only the object mask.

### 2.2 Light pattern design

In addition to the One-Light-At-Time (OLAT) pattern, we have carefully designed 13 different light patterns for our dataset. These patterns involve lighting multiple LED lights either randomly or in a regular manner.

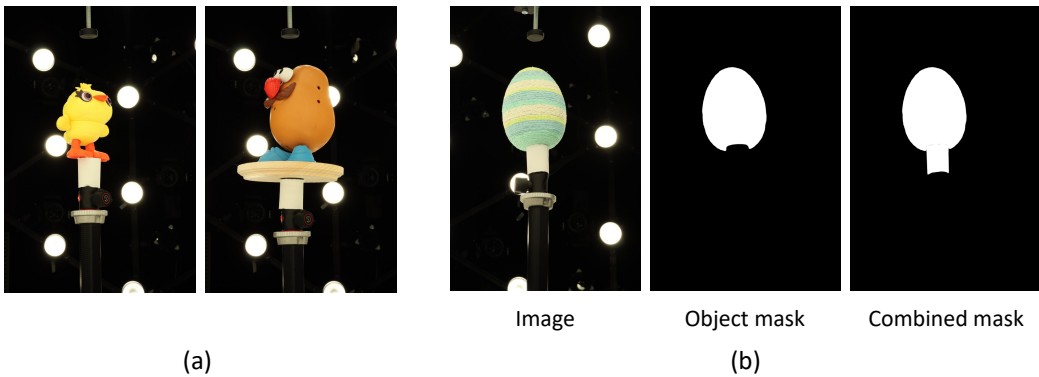

|   |   |
|---|---|
| Image | Object mask | Combined mask |

(a)                                    (b)

Figure 1: **(a)** Two types of supporting devices used in our dataset. **(b)** We use the combined masks for training to eliminate density ambiguity.

For the first 6 light patterns (001 to 006), we divide the 142 lights into 6 groups based on their spatial location. Each light pattern corresponds to activating one of these groups.

As for the remaining 7 light patterns (007 to 013), the lights are randomly illuminated, with the total number of chosen lights gradually increasing.

Fig. 2 illustrates the 13 light patterns present in our dataset.

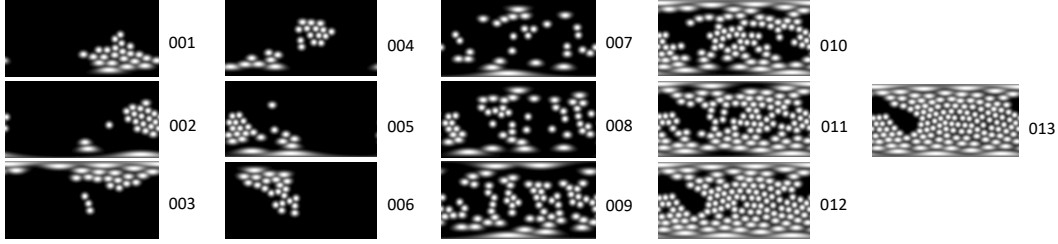

Figure 2: 13 kinds of light patterns in our dataset, shown as an environment map.

## 2.3 Chrome Ball

In order to perform light calibration, we need to determine the radius and center of the chrome ball in the world coordinate system. This information is crucial for calculating the surface normals at each point on the ball's surface. To ensure accurate intersection point computation, it is important to obtain the radius and position of the chrome ball on the same scale as the camera poses.

To achieve this, we propose using NeuS [3] to extract a mesh with a scale matching the camera poses. We provide multi-view images of the mirror ball as input to NeuS. However, since the mirror ball is highly reflective and difficult to reconstruct accurately using NeuS, we fill the foreground pixels of the mirror ball with black.

Finally, we fit a sphere to the extracted mesh to determine the location and radius of the mirror ball, which allows us to obtain the necessary information for light calibration.

## 2.4 Image Resolution

The image resolution we use for capturing is $2656 \times 3984$. For novel view synthesis experiments, we use half of the resolution, i.e., $1328 \times 1992$. For inverse rendering experiments, we use $800 \times 1200$ resolution since the inverse rendering methods are typically time-consuming.

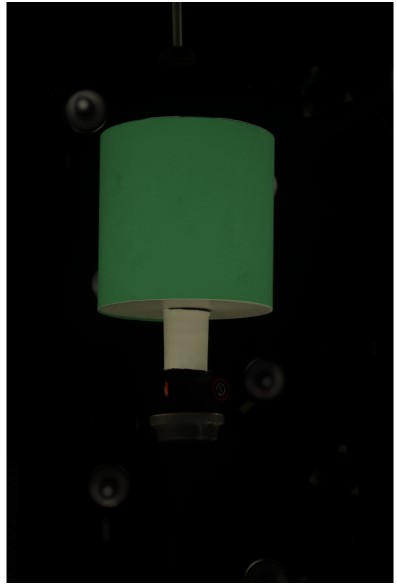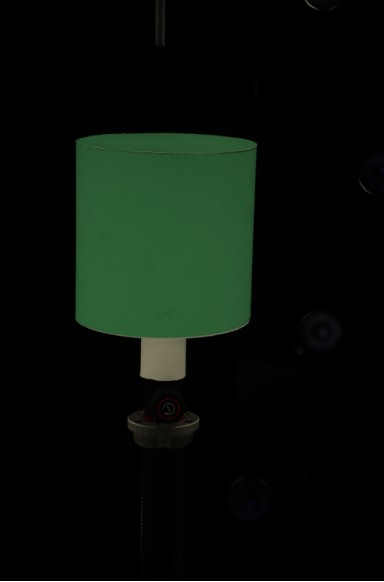

Figure 3: **Example images of the cylinder.**

## 2.5 Camera parameters

During capturing, we set the camera ISO to 100, aperture to F16, and shutter speed to 1/5. We use Daylight mode for its white balance.

We did not perform extra color calibration for the same type of cameras. While it's acknowledged that certain inherent camera intrinsic differences and uncontrollable variables may result in occasional color differences, we can observe that the potential differences are very small and negligible from the images and the experimental results.

To further quantify the differences between different cameras, we designed a small experiment. We captured a 3D-printed cylinder, covered with a type of diffuse green paper. The visualization is in Fig. 3. The basic idea is to compute the difference in object surface colors across different cameras. This calculation serves as a rough measurement of the intrinsic differences among different cameras.

To reduce the impact of specular reflections, we use polarizers on the camera systems. In addition, we selected adjacent cameras to reduce the influence of view-dependent color variations. Our findings indicate that the differences between different cameras amount to approximately 1%.

As a result, we can observe that cameras of the same type after setting the same camera parameters already exhibit a high level of consistency without supplementary post-processing calibration procedures.

## 3 More details of evaluation results

### 3.1 Code to reproduce the results in the paper

We use the open-source code repositories for the baselines in the paper.

- **NeRD**: https://github.com/cgtuebingen/NeRD-Neural-Reflectance-Decomposition
- **Neural-PIL**: https://github.com/cgtuebingen/Neural-PIL
- **PhySG**: https://github.com/Kai-46/PhySG
- **InvRender**: https://github.com/zju3dv/InvRender
- **Nvdiffrec-mc**: https://github.com/NVlabs/nvdiffrecmc
- **TensoIR**: https://github.com/Haian-Jin/TensoIR

- **NeRF**: https://github.com/KAIR-BAIR/nerfacc
- **TensoRF**: https://github.com/apchenstu/TensoRF
- **instant-NGP**: https://github.com/bennyguo/instant-nsr-pl
- **NeuS**: https://github.com/bennyguo/instant-nsr-pl

## 3.2 Computational resources

We use a single GTX 2080 GPU for each object to run the baseline experiments.

## 3.3 Relighting evaluation

We conducted an evaluation of all 64 objects in our dataset using TensoIR [1], which is one of the most recent state-of-the-art (SOTA) inverse rendering methods capable of multi-illumination optimization. For each object, we evaluated the performance of TensoIR under single illumination, multi-illumination, and relighting using novel illuminations. The evaluation results can be found in Tab. 1. Additionally, we include visualizations of the results for a selected number of objects in Fig. 4. As mentioned in the main paper, our dataset provides ground-truth information for the 142 linear polarized LED lights. This allows for the quantitative evaluation of the relighting quality. However, comparing the relighting results directly with the captures without aligning the albedo or light intensity between the two is impractical due to the ambiguity between them in the rendering equation. In practice, we train TensoIR under three different light patterns given their corresponding ground-truth illumination. During the evaluation, we used a different set of ground-truth illumination, along with the learned object's geometry and BRDF, to relight the object. We then compared the relit images with the captures under the new illumination to obtain our relighting evaluation metrics.

Tab. 1 presents the quantitative results of TensoIR's relighting performance on all 64 objects with various materials in our dataset. We used light patterns *009*, *011*, and *013* for training, and the remaining light patterns for evaluation.

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

| Object ID | Material | Single Illum | Multi-Illum | Relighting | Object ID | Material | Single Illum | Multi-Illum | Relighting |
|---|---|---|---|---|---|---|---|---|---|
| 1 | plastic | 24.43 | 24.70 | 26.41 | 33 | fabric | 29.30 | 28.97 | 28.00 |
| 2 | paper | 34.13 | 34.48 | 31.99 | 34 | wax | 43.36 | 42.39 | 34.29 |
| 3 | plastic | 36.21 | 35.84 | 33.01 | 35 | clear-plastic | 22.53 | 22.58 | 22.96 |
| 4 | stone | 31.15 | 30.07 | 31.07 | 36 | sponge | 32.49 | 32.18 | 30.86 |
| 5 | painted | 30.53 | 30.80 | 30.16 | 37 | fabric | 30.98 | 30.58 | 28.70 |
| 6 | ceramic | 35.49 | 35.40 | 33.07 | 38 | foliage | 25.53 | 25.03 | 26.89 |
| 7 | fabric | 32.64 | 31.87 | 29.65 | 39 | plastic | 34.04 | 33.67 | 31.46 |
| 8 | clear-plastic | 23.12 | 23.43 | 26.41 | 40 | plant | 27.74 | 27.61 | 30.26 |
| 9 | paper | 33.98 | 33.53 | 31.21 | 41 | fabric | 34.12 | 33.66 | 29.51 |
| 10 | paper, plastic | 30.15 | 29.92 | 28.66 | 42 | food | 35.18 | 34.69 | 32.13 |
| 11 | plastic | 34.19 | 33.81 | 29.95 | 43 | paper | 42.16 | 41.26 | 31.29 |
| 12 | leather | 28.89 | 28.45 | 29.47 | 44 | fabric | 32.78 | 32.38 | 29.45 |
| 13 | ceramic | 32.04 | 32.16 | 29.95 | 45 | metal | 28.22 | 28.08 | 29.62 |
| 14 | metal, plastic | 27.82 | 28.12 | 29.31 | 46 | fabric | 29.02 | 28.59 | 29.58 |
| 15 | fabric, plastic | 28.99 | 28.78 | 28.71 | 47 | painted | 32.16 | 31.71 | 33.62 |
| 16 | plastic | 35.02 | 34.70 | 32.84 | 48 | metal | 29.09 | 29.57 | 27.13 |
| 17 | coated | 26.18 | 26.52 | 27.57 | 49 | rubbery | 27.51 | 27.22 | 28.69 |
| 18 | glass | 29.61 | 29.54 | 27.68 | 50 | fabric | 31.31 | 30.55 | 28.66 |
| 19 | ceramic | 31.56 | 31.22 | 29.55 | 51 | plastic | 30.47 | 29.90 | 30.56 |
| 20 | ceramic | 29.31 | 29.31 | 28.63 | 52 | hair | 22.65 | 22.50 | 22.51 |
| 21 | paper | 35.94 | 35.39 | 29.85 | 53 | rubbery | 30.21 | 30.77 | 28.32 |
| 22 | wooden | 19.72 | 20.30 | 23.00 | 54 | leather | 29.60 | 29.34 | 29.84 |
| 23 | paper | 36.18 | 35.46 | 30.82 | 55 | stone | 36.33 | 35.79 | 31.99 |
| 24 | latex | 26.22 | 26.04 | 27.32 | 56 | fabric | 30.21 | 30.12 | 29.03 |
| 25 | latex | 28.93 | 28.67 | 27.93 | 57 | cloth | 30.05 | 29.77 | 24.70 |
| 26 | wicker | 28.97 | 28.26 | 27.16 | 58 | wicker | 28.76 | 28.45 | 28.77 |
| 27 | foam | 34.03 | 33.36 | 30.70 | 59 | nylon | 34.70 | 34.52 | 32.75 |
| 28 | metal | 28.82 | 28.57 | 30.82 | 60 | fabric | 36.66 | 36.13 | 33.61 |
| 29 | fabric | 32.30 | 31.83 | 32.38 | 61 | fabric | 29.93 | 29.51 | 29.35 |
| 30 | foam | 30.20 | 29.97 | 29.97 | 62 | fabric | 36.56 | 35.90 | 31.85 |
| 31 | painted | 30.66 | 30.58 | 30.00 | 63 | fabric | 35.63 | 35.53 | 31.95 |
| 32 | stone | 31.82 | 31.53 | 29.75 | 64 | paper | 31.24 | 30.01 | 27.62 |

Table 1: **Evaluation results of TensoIR on all objects in our dataset**. We report the PSNR values of each object under single illumination, multi-illumination, and their relighting PSNR under novel illuminations.

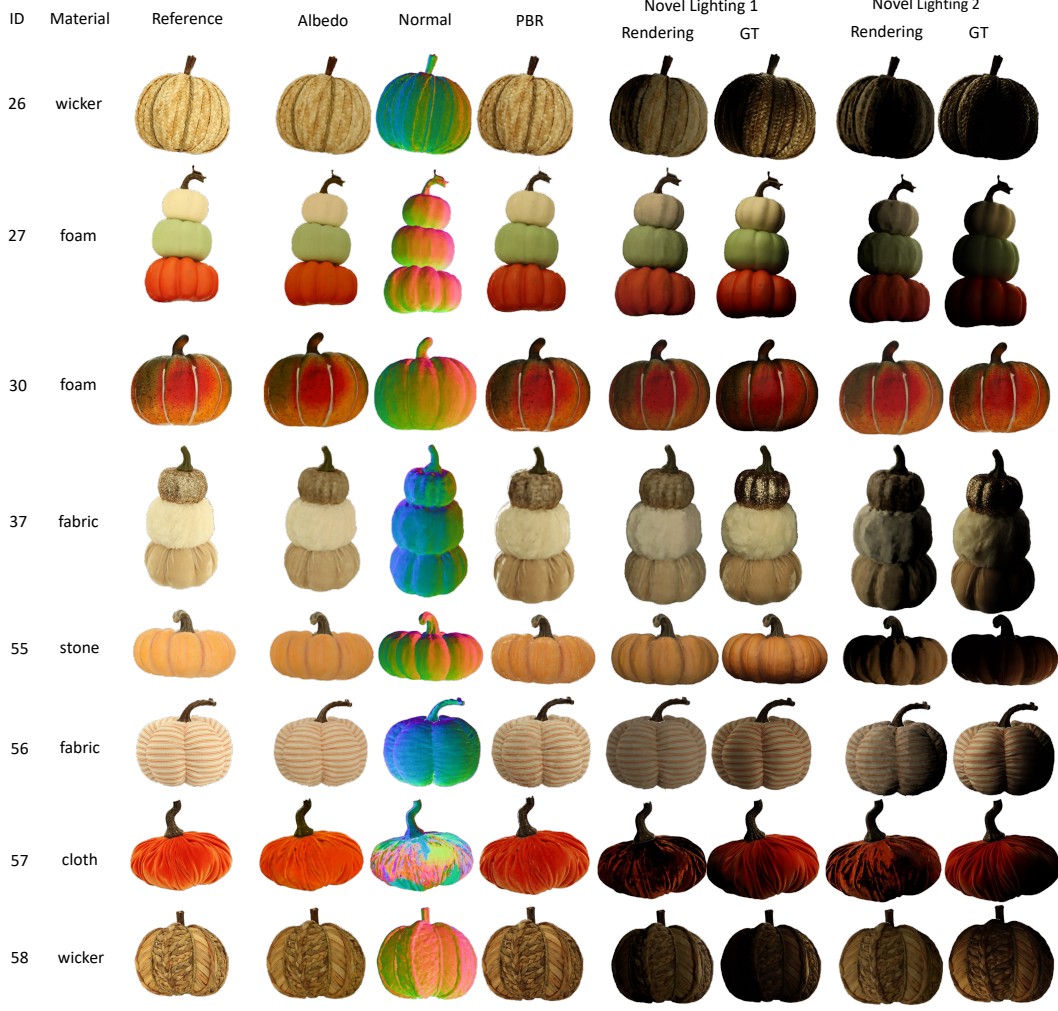

Figure 4: **Material reconstruction and relighting results on a selective number of objects in our dataset.** We show the decomposed albedo, normal, rendering image, and relighting image under novel illumination. In general, objects with diffuse surfaces have better results than objects with specular surfaces. For example, it is difficult to correctly reconstruct normal in highly-specular areas for object No.37.