# OpenReview forum: "OpenIllumination: A Multi-Illumination Dataset for Inverse Rendering Evaluation on Real Objects"
_NeurIPS.cc/2023/Track/Datasets_and_Benchmarks — NeurIPS 2023 Datasets and Benchmarks Poster_

### Official Review · Reviewer_dCTX · 2023-07-21
**Real-world data for relighting evaluation captured under ideal conditions**

**Rating:** 6
**Confidence:** 4
**Clarity:** The paper is easy to read.

**Strengths:**

- The light stage setup reduces acquisition time resulting in a dataset with many objects and materials.

- The light patterns used are more complex compared to other datasets like ReNe or DTU.

- Lights are represented as spherical Gaussians which make them easy to use in methods that use the same representation for the illumination.

- The light calibration using the chrome ball is interesting.

- The experiments demonstrate the use of the dataset for evaluation of inverse rendering methods.

**Additional Feedback:**

The dataset and the paper are OK but the lab setting is not very exciting.
The high PSNR values in Table 4 and the example in Figure 6 show that the noise in the dataset is very small as current methods perform well. It can also mean that the dataset is already too easy for some methods.

**Correctness:**

The dataset construction seems sound but there are some open questions.
- Is the choice to use a global lobe parameter based on data showing a very small variance?
- Were the cameras calibrated in any way to ensure that images from different cameras show the same colors?

**Documentation:**

The dataset page on Huggingface provides a suitable description of the data.

**Ethics:**

The dataset contains no personal data or offending images to my understanding. No additional review needed.

**Limitations:**

A discussion of limitations is missing. The images of the objects are taken in an ideal lab environment. It is necessary to discuss possible disadvantages of this. See also the comments about possible improvements.


**Opportunities For Improvement:**

- While the light stage setup is very efficient it is also limited to lab environments. Many methods and researchers aim to reconstruct shape and materials from images taken in the wild with commodity cameras to make reconstruction cheap and accessible. The paper uses a close to ideal capture environment trying to make it as easy as possible for the inverse rendering methods, which could even hamper development of robust methods. The paper describes this in ll.22-25 and section 4.5 where the photographer and the camera become annoyances that should be avoided to not interfere with the assumptions of the methods. Capturing each of the objects with a hand-held camera in random everyday environments as input data and using the presented dataset as the ground truth data would add a lot of additional value and use cases.

- There are no experiments with synthetic data to show the advantages of the real data. Synthetic data is a lot easier to produce. Adding experiments on synthetic data could show how important testing on real data is.

- l. 45 is not clear. 30 FPS is not much. Are the DSLRs considered to be high-speed cameras?

**Relation To Prior Work:**

Some references are missing. The DTU dataset could be briefly mentioned even if it has a different focus because it also uses light patterns.
Please also consider to add
- Rudnev et al., "NeRF for Outdoor Scene Relighting" ECCV 2022
- Shi et al., "A Benchmark Dataset and Evaluation for Non-Lambertian and Uncalibrated Photometric Stereo" CVPR 2016

**Summary And Contributions:**

The paper presents a dataset with images of objects taken under multiple known illumination conditions.
It allows to evaluate the performance of inverse rendering methods for relighting and novel view synthesis.
This is important because real data for evaluating relighting is scarce.
The data acquisition is optimized for speed rather than realistic conditions and uses a light stage with fixed light and camera positions.

---

> ### Author Response · Authors · 2023-08-21
>
> **Lab or in-the-wild environment.**
> We would like to emphasize the major purpose of introducing this dataset is to provide ground-truth information, such as relit images under different illumination, for real-world objects. Previous inverse rendering works usually use commodity cameras to capture in-the-wild images to validate their methods. However, it is much more difficult for them to obtain the ground truth. This would end up just being images without much use for inverse rendering. Moreover, given that our dataset is acquired in a light stage, it is easy to virtually relight and change view to simulate uncontrolled captures in general illuminations.
>
>
> **Synthetic vs. real data.**
> While synthetic data is easy to produce, our real objects dataset with much ground truth information has the following two advantages.
> First, real captured images have much better image quality than synthetic data. And the materials of many real objects have more complex visual properties than synthetic data, such as the shining plastic cup in the 5th and 10th row of Fig. 3, which is hard to simulate in rendering engines.
> Second, Most commonly used synthetic data are rendered with a physically-based rendering model, such as the Disney BRDF model, and most inverse rendering work also uses similar rendering models to achieve physically-based rendering reconstruction. If both the data and method use the same rendering model, the method will attain a good reconstruction quality more easily. However, the reconstruction quality may drop greatly on synthetic data rendered with another rendering model or real-captured images. Our real dataset with ground truth relighting information can be used as a useful evaluation dataset to testify each method.
>
>
> **Camera speed.**
> We would like to clarify that the fps of cameras correspond to burst mode instead of video mode. DSLR cameras only have 5 fps under burst mode.
>
> **Limitations.**
> As suggested, we have added a section to discuss the limitations of our work. Please refer to Sec. 5 in the main paper.
>
> **Lobe parameter.**
> Since the lights share identical type and intensity, the lobe parameter remains consistent across all the lights. We have added this explanation in the paper.
>
> **About ensuring images show the same color from different cameras.**
> We have two types of cameras in our dataset: DSLR cameras and high-speed cameras. As described in the supplementary materials, we have calibrated the parameters of the cameras of the same type to be consistent, including ISO, aperture, and shutter speed to ensure the images show the same colors.
>
> **Prior work.**
> We have added the comparison against the three papers mentioned by the reviewer. Please refer to Tab. 1 in the revised main paper.
>
> **High PSNR.**
> We would like to clarify that the proposed dataset is actually not very easy for current methods. First, as shown in Tab. 2 to Tab. 5 in the revised paper, there are still about half of the objects with a PSNR lower than 30. Second, there are some objects in the dataset, such as the bucket and the plastic cup (4th and 5th rows in Fig. 3), that all the methods struggle to achieve a good reconstruction quality.

---

> > ### Comment · Reviewer_dCTX · 2023-08-27
> >
> > Thank you for your response.
> >
> > I could not find the information about color calibration in the supplement.
> > Are the parameters that get calibrated just the settings that can be set on the camera? (ISO, aperture, shutter, WB)

---

> > > ### Author Response · Authors · 2023-08-29
> > >
> > > We have tried our best to enhance color consistency across various images in the relighting dataset. We use the same type of camera and manually set the controllable camera parameters to be the same, including ISO, aperture, shutter speed, and white balance for every camera. While it's acknowledged that certain inherent camera intrinsic differences and uncontrollable variables may result in occasional color differences, we can observe that the potential differences are very small and negligible from the images and the experimental results.

---

> > > > ### Comment · Reviewer_dCTX · 2023-08-29
> > > >
> > > > I understand that there has been no color calibration among the different cameras (of same type).
> > > > How did you quantify the potential differences?
> > > > It would be valuable to add this information to the paper.

---

> > > > > ### Author Response · Authors · 2023-08-29
> > > > >
> > > > > Thank you for your valuable suggestion. We have incorporated your input by introducing a comprehensive discussion in the supplementary section. Please refer to Sec. 2.4 in the supplementary.
> > > > >
> > > > > In our study, we captured a 3D-printed cylinder, covered with a type of diffuse green paper. The basic idea is to compute the difference in object surface colors across different cameras. This calculation serves as a rough measurement of the intrinsic differences among different cameras.
> > > > >
> > > > > To reduce the impact of specular reflections, we use polarizers on the camera systems. In addition, we selected adjacent cameras to reduce the influence of view-dependent color variations. Our findings indicate that the differences between different cameras amount to approximately 1%.
> > > > >
> > > > > As a result, we can observe that cameras of the same type after setting the same camera parameters already exhibit a high level of consistency without supplementary post-processing calibration procedures.

---

> > > > > > ### Comment · Reviewer_dCTX · 2023-08-29
> > > > > >
> > > > > > Thank you for adding this

---

### Official Review · Reviewer_rUYo · 2023-07-21

**Rating:** 7
**Confidence:** 4
**Correctness:** The dataset is constructed in a sound…

**Strengths:**

A large number of carefully calibrated images with controlled lighting of objects.

Baseline experiments for several representative tasks/applications.

**Additional Feedback:**

It would be helpful to justify the settings used in the experiments, e.g. number of lights, lighting patterns, number of cameras, etc. and clarify what will be needed/sufficient for different experiments.

**Clarity:**

The paper is generally clear, but there are a few typos, e.g. in the title "Dateset" should be "Dataset".

**Documentation:**

Most details are provided; some decisions (such as objects to be included etc.) should be further clarified.

**Ethics:**

No.

**Limitations:**

Limitations of the dataset/work should be discussed explicitly.

**Opportunities For Improvement:**

The dataset only contains a small number of objects, but it is already much larger than existing datasets of a similar nature.

Some justifications could be provided regarding the choice of objects in the dataset.

**Relation To Prior Work:**

Relations to prior work are clearly presented, highlighting the unique benefits of the proposed dataset.

**Summary And Contributions:**

This paper presents a dataset containing a large number of images captured by a light stage, with accurate calibration, which enables a range of applications. The paper includes baseline experimental results for several tasks.

---

> ### Author Response · Authors · 2023-08-21
>
> **Dataset size.**
> We agree with the reviewer that the number of objects in the proposed dataset is relatively small. However, as shown in Tab. 1, our dataset has been larger than previous inverse rendering datasets. Moreover, our dataset ensures a large number of different materials, views, and illuminations simultaneously, which has not been achieved at a competitive scale in previous works.
>
>
> **Choice of objects.**
> We are selecting objects to cover a wide range of different materials. The figure on page 4 of the main paper shows the distribution of the object materials in the dataset. For the objects belonging to one material, we try to select objects with different shapes and appearances.
>
> **Limitations**
> As suggested, we have added a section to discuss the limitations of our work. Please see Sec. 5 in the revised paper.
>
> **Typos**
> As suggested, we have carefully refined the paper by correcting the typos.
>
> **Experiment settings.**
> We have explained the settings used in the experiments in the paper. Please refer to Sec. 4. in the main paper and Sec. 3.2 in the supplementary material. For all the novel view synthesis and inverse rendering experiments, we randomly split the images into 38 views for training and 10 views for testing. For the novel-view-synthesis and single illumination inverse rendering experiments, we used images corresponding to light pattern 013. For the multi-illumination inverse rendering experiments, we use light patterns 009, 011, and 013.

---

### Official Review · Reviewer_USxp · 2023-07-29
**A good dataset for inverse rendering and novel view synthesis benchmarking, but paper exposition is improvable.**

**Rating:** 5
**Confidence:** 4

**Strengths:**

+ Real-world datasets with ground truth are typically difficult and expensive to capture. The are important to validate methods which were trained on synthetic or real data.
+ The dataset has a reasonable size

**Additional Feedback:**

Minor comments:

- L45 states "high-speed cameras running at 30 fps", but any camera can run 30Hz. So why is a high-speed camera needed?

- L45: OLAT is used the first time here, but defined only much later in L89.

- L61: An intention to release data and code is not a separate contribution. The entire paper is meaningless without a release.

- L96: "large" -> "larger"

- Eq.(3): It is better to be more precise and state what distance is used for d( , ) and which norm is used for |p|

- L155 states that SAM performance is unsatisfactory and later in L158 it is stated the segmentation is performed with multiple bounding box prompts. Here, it is not clear whether segmentation still refers to SAM and how the multiple prompts are generated.

- L223: "recosntruct 3D surface" -> "reconstruct a 3D surface"

**Clarity:**

The paper is mostly well-written and structured.
Minor comments on the writing are provided below.

**Correctness:**

The proposed dataset is well constructed.

One slightly worrying aspect mentioned in the supplementary material is that the dataset publication still needs approval from the company that was involved in the capturing process.

**Documentation:**

I have checked the landing page for the data download, which contains plenty of useful information and documentation.
Hosting, licensing and maintenance plan seem to well sorted out.

**Ethics:**

I do not see ethical concerns with this submission.

**Limitations:**

There is no separate discussion of limitations which would be a nice addition to the paper, e.g. potential domain gaps with respect to in the wild scenarios.

Negative societal impact is not discussed, but I do not see any risk of negative impact.


**Opportunities For Improvement:**

- The experiment in Fig. 6 is rather meaningless without details about the hand-held camera reconstruction, e.g. what kind of camera was used? Are the image resolution, camera distance similar to the ones in the camera dome?
How many views were used to reconstruct this example? Which reconstruction method was used?

- For the experiment in Table 4, I recommend adding some more baselines, e.g. classic NeRF, NeRD, Physg, Munkberg et al. CVPR'22. There is a large body of NeRF-related works and currently the selection of works seems minimal (just two) and a bit arbitrary.

Further baselines (and related work) could include:

[A] Jon Hasselgren, Nikolai Hofmann, and Jacob Munkberg. Shape, light & material decomposition from images using Monte Carlo rendering and denoising. In NeurIPS, 2022.

[B] Haoqian Wu, Zhipeng Hu, Lincheng Li, Yongqiang Zhang, Changjie Fan, and Xin Yu. NeFII: Inverse rendering for reflectance decomposition with near-field indirect illumination. In CVPR, 2023.



- Add missing details to the paper: which image resolution was used for capturing and which for the benchmark evaluation?

- The paper does not seem to include the standard checklist https://neurips.cc/public/guides/PaperChecklist nor the recommended data sheet for new datasets, e.g. https://arxiv.org/pdf/1803.09010.pdf
This is a crucial addition to make potentially raise my evaluation score.


**Relation To Prior Work:**


Relations to prior work are mostly well described. Nevertheless, the overview of existing works (Tab. 1) could be much more comprehensive, e.g.:


[C] Roger Grosse, Micah K Johnson, Edward H Adelson, and William T Freeman. Ground truth dataset and baseline evaluations for intrinsic image algorithms. In ICCV, 2009

[D] ShapeNet Intrinsics: Jian Shi, Yue Dong, Hao Su, and Stella X Yu. Learning non-Lambertian object intrinsics across ShapeNet
categories. In CVPR, 2017

[E] Min Li, Zhenglong Zhou, Zhe Wu, Boxin Shi, Changyu Diao, and Ping Tan. Multi-view photometric stereo: A robust solution and benchmark dataset for spatially varying isotropic materials. IEEE TIP, 29:4159–4173, 2020


[F] Zhengfei Kuang, Kyle Olszewski, Menglei Chai, Zeng Huang, Panos Achlioptas, and Sergey Tulyakov. NeROIC: neural rendering of objects from online image collections. ACM TOG, 41(4):1–12, 2022.


[G] Matt Deitke, Dustin Schwenk, Jordi Salvador, Luca Weihs, Oscar Michel, Eli VanderBilt, Ludwig Schmidt, Kiana Ehsani, Aniruddha Kembhavi, and Ali Farhadi. Objaverse: A universe of annotated 3D objects. In CVPR, 2023



**Summary And Contributions:**

The paper presents a new real-world dataset for benchmarking inverse rendering and novel view synthesis methods.
The datasets consists of 64 different objects which were captured simultaneously by 72 cameras in a half-dome setup under various controlled light patterns.

---

> ### Author Response · Authors · 2023-08-21
>
> **Details of experiment in Fig.6.**
> We use a smartphone camera to capture the images. The image resolution is 1920x1080. The distance between the camera and the object is about 0.5m, which is smaller than the distance in the dome setup. We used TensoIR to reconstruct this example with 99 training views and 25 testing views. We would like to note that the main point we are explaining in this experiment is that the handheld camera setup often fails to ensure full-view coverage and consistent illumination. Thus, it is hard to know if the incomplete reconstruction is attributed to the low-quality data or the inferior ability of the method itself. In contrast, our dataset delivers a consistent and high view coverage, thus being more reliable for inverse rendering evaluation.
>
>
> **More baselines.**
> In Tab. 4 (Tab. 5 in the revised paper), we are comparing novel view synthesis methods, not inverse rendering methods. We have added vanilla NeRF and instant-NGP into the experiments. Additionally, we have added PhySG, InvRender, and nvdiffrec-mc to inverse rendering experiments in Tab. 2.
>
> **Image resolutions.**
> The image resolution for capturing is 2656 $\times$ 3984. For novel view synthesis, we use half of the resolution, i.e., 1328 $\times$ 1992. For inverse rendering, we use 800$\times$ 1200 resolution since the inverse rendering methods are typically time-consuming. We have added the details in the revised paper.
>
> **Checklist.**
> We have added the checklist and data sheet to the revised paper. Please refer to the Checklist section right after the references.
>
>
> **Limitation discussion.**
> As suggested, we have added a section to discuss the limitations of our work. Please refer to Sec. 5 in the revised paper.
>
> **Societal impact.**
> There is not any negative societal impact. We have added a discussion in the revised paper. Please refer to the Checklist section.
>
> **Camera speed.**
> We would like to clarify that the fps of cameras correspond to burst mode instead of video mode.
>
> **Release data and code should not be a separate contribution.**
> As suggested, we have removed the corresponding sentence in the revised paper.
>
> **Relation To Prior Work**
> As suggested, we have added three works of the mentioned datasets to Tab. 1. We thank the reviewer for providing them to make the comparison much more comprehensive.
> Moreover, we would like to clarify that the rest two of them (ShapeNet intrinsics and Objaverse) are not real data. Thus, we didn’t include them in Tab. 1 since we are comparing real data.
>
>
> **Segmentation.**
> Yes, the segmentation still refers to SAM. The original SAM only supports a single bounding box prompt and multiple points prompts.  In practice, we find that the segmentation performance is unsatisfactory when using the single bounding box prompt since it usually over-segments the object and fails to segment very detailed structures accurately. Thus, we use multiple bounding box prompts to segment the object to increase the segmentation performance. The multiple bounding box prompts are generated by manually labeling.
>
> **Distance norm.**
> In Eq.(3), the distance $d$ and norm $\|p\|$ are both L2 norm. We have made this clear in the revised paper.
>
> **Other minor comments.**
> As suggested, we have carefully refined the paper by correcting the typos and adding some detailed explanations.

---

### Author Response · Authors · 2023-08-21

Dear Reviewers,

Thank you for dedicating your time to review our paper and offering insightful feedback. We sincerely appreciate your efforts to help enhance the quality of our research.
We are pleased to note that all reviewers were supportive of our work for recognizing our contribution in proposing a dataset that is real-world (USxp), large-size (USxp and rUYo), contains ground-truth information that is difficult and expensive to obtain (USxp, rUYo, and dCTX), and plays an important role in validating inverse rendering methods (USxp and dCTX).

We have made a revision to our main paper and supplementary material. The major changes are 1) Adding a section to discuss limitations; 2) Adding a checklist section; 3) Adding more related datasets in comparison; 4) Adding more baseline experiments. We would like to thank to all the reviewers again for helping enhance the quality of our work.

We are also pleased to inform you that the release of the dataset is nearly completed. The company has approved making everything public, including the dataset, project page, and code. We will inform you as soon as possible if the dataset is released before the end of the author-reviewer discussion.

---

### Decision · Program_Chairs · 2023-09-22

**Decision:**

Accept (Poster)

**Comment:**

As all reviewers agreed, the quality of the dataset was acceptable and the dataset was well structured. The main concern is that the dataset needs to be approved by the company before it can be made public. Since the author states that the company has approved the public content, although it is not yet published, it is inclined to believe that the author's promise is true. Therefore, the conclusion of acceptance is given.